# Node Monitoring as a Fault Detection Countermeasure against Information Leakage within a RISC-V Microprocessor

Donald E. Owen, Jr. [1], Jithin Joseph [2], Jim Plusquellic [2,*], Tom J. Mannos [1] and Brian Dziki [3]

1 Advanced CMOS Products/Design, Sandia National Laboratories, Albuquerque, NM 87131, USA; deowen@sandia.gov (D.E.O.J.); tjmanno@sandia.gov (T.J.M.)
2 Electrical and Computer Engineering, University of New Mexico, Albuquerque, NM 87131, USA; jithinjoseph@unm.edu
3 Information Assurance Research, Department of Defense, Fort G. G. Meade, Anne Arundel County, MD 24003, USA; bjdziki@tycho.ncsc.mil
* Correspondence: jimp@ece.unm.edu

**Abstract:** Advanced, superscalar microprocessors ($\mu P$) are highly susceptible to wear-out failures because of their highly complex, densely packed circuit structure and extreme operational frequencies. Although many types of fault detection and mitigation strategies have been proposed, none have addressed the specific problem of detecting faults that lead to information leakage events on I/O channels of the $\mu P$. Information leakage can be defined very generally as any type of output that the executing program did not intend to produce. In this work, we restrict this definition to output that represents a security concern, and in particular, to the leakage of plaintext or encryption keys, and propose a counter-based countermeasure to detect faults that cause this type of leakage event. Fault injection (FI) experiments are carried out on two RISC-V microprocessors emulated as soft cores on a Xilinx multi-processor System-on-chip (MPSoC) FPGA. The $\mu P$ designs are instrumented with a set of counters that records the number of transitions that occur on internal nodes. The transition counts are collected from all internal nodes under both fault-free and faulty conditions, and are analyzed to determine which counters provide the highest fault coverage and lowest latency for detecting leakage faults. We show that complete coverage of all leakage faults is possible using only a single counter strategically placed within the branch compare logic of the $\mu Ps$.

**Keywords:** information leakage; RISC-V; FPGA fault injection and emulation

## 1. Introduction

Fail secure refers to systems that incorporate countermeasures (CMs) that prevent sensitive data from being exposed, e.g., on I/O channels, when a fault occurs. In the context of a microprocessor ($\mu P$), applications that encrypt data are very common, and fall into a special class of applications that possess private information that should not be revealed to outside entities, namely, plaintexts and encryption keys. Fail secure in this context refers to $\mu Ps$ outfitted with CMs that detect and prevent faults that would otherwise leak keys and plaintexts through I/O channels. Note that CMs which address private information leakage can be designed to target only those faults that lead to leakage events, and as a result, can have a smaller negative impact on the performance and area of the $\mu P$.

There exists a wide variety of software and hardware CMs that have been proposed to provide fault tolerance within the complex system architectures of $\mu Ps$ [1]. All fault tolerant techniques require a detection mechanism and then some type of mitigation strategy. Many of the previously proposed techniques introduce redundancy and voting to detect and then correct fault effects, but in general, these approaches are considered heavy-weight because of the adverse impact they have on the size of the hardware implementation and its performance [2,3]. Hardware-software-based continuous symptom monitoring strategies are lighter weight but require moderate to significant changes to the $\mu P$ architecture [4,5].

The latency of software monitors that leverage existing hardware features may be too large to prevent leakage [6]. Similarly, on-line techniques introduce specialized instructions within the hardware-software stack of the $\mu P$ and carry out periodic built-in self-test using a scan chain infrastructure [7].

In contrast, our proposed periodic testing method introduces only a small set of counters and halts execution on the activation of information leakage faults (and on a large subset of non-leakage fault types). Although the proposed countermeasure is able to detect different types of leakage sensitive faults, our primary goal is to detect delay faults because they represent a precursor to hard faults, e.g., stuck at 0 and stuck at 1. Given most $\mu Ps$ enable frequency control, one possible approach of applying the proposed periodic testing strategy is to test a counter-instrumented RISC-V design at the highest possible frequency in an attempt to detect a delay fault from a set that have been pre-identified as leading to a fail-insecure outcome.

In this work, we evaluate the counter-based CM on the Rocket $\mu P$ [8] by executing the Advanced Encryption Algorithm (AES) with a fixed plaintext and key in a large set of proof-of-concept FPGA emulation experiments. Although the counter values depend on the specific plaintext and key used in the encryption operation, our previous work [9] shows a large overlap exists in the sets of leakage sensitive faults in two different plaintext/key combinations, which suggests the proposed strategy is data insensitive and will remain effective for other plaintext/key combinations. As further support of the approach, a second $\mu P$, called Potato [10], is tested with the counter-based CMs and the results are shown to be nearly identical.

Although the simplest mitigation strategy after detecting the occurrence of a leakage sensitive fault is simply to halt execution of the processor, other mitigation strategies described in previous work, e.g., checkpoint and replay, are also possible. For these more complex recovery mechanisms, and especially for fail-operational designs where execution needs to continue in spite of failure, the latency between fault occurrence and fault detection becomes an important factor in reducing the complexity and effectiveness of the recovery mechanism.

To address this issue, we also investigate the latency of the counter-based CM. Although the latency analysis is relevant only to continuous symptom monitoring-based methods, our findings reveal that increasing the number of counters decreases the latency of detecting faults and therefore, counter set size and latency represent a trade-off. Given the ease at which counters can be added to the design, we argue that the ability to tune the number of counters to meet the requirements of the recovery mitigation strategy is an attractive feature. Moreover, the nodes identified by the counter analysis presented here are highly sensitive to fault propagation effects, and therefore, they represent attractive targets for alternative continuous symptom monitor-based approaches.

*Contributions*

In this paper, we utilize the fault injection engine and instrumented RISC-V Rocket design [11] from previous work as well as the post-processing system developed for identifying leakage faults [9,12,13]. The counter-based CM, on the other hand, is a novel contribution of this paper. The specific contributions of this work include:

- The evaluation of a counter-based CM that can detect leakage sensitive faults (also called severe) with latencies sufficiently low to prevent leakage in a periodic testing paradigm.
- The identification of a set of counters that provide the highest fault coverage and lowest latency for the targeted information leakage faults.
- An extended analysis that reveals the collateral fault coverage provided by the selected counters for detecting faults from the larger sets of 85,714 stuck at 0, stuck at 1, delay and invert faults.
- An enhancement to a codesign-based fault emulation engine developed in previous work that enables the latencies of fault effects to be determined quickly using a binary search algorithm.

Although the experiments are conducted as emulations on an FPGA, the proposed CM is designed to be integrated into a hard-wired, custom implementation of the $\mu Ps$. The synthesis tool flow which converts the designs to an FPGA implementation is constrained to ensure the results presented in this paper are applicable to hard-wire versions.

The remainder of this paper is organized as follows. Section 2 discusses additional related work. Section 3 describes the experimental design and attributes of the FI experiments. Section 4 describes the fault coverage results obtained using a subset of counters while Section 4.3 presents an analysis of latency. Section 5 presents our conclusions and future plans.

## 2. Related Work

The previous work presented in this section provides additional references and details on the techniques described in the Introduction. Although the related work described here addresses fault tolerance in $\mu P$, to the best of our knowledge, no previous work exists that is focused on the detection of leakage sensitive faults.

A transient fault detection continuous symptom monitor (CSM) technique is proposed in [4] in which a program is duplicated and run concurrently as multiple threads on a processor. It leverages a microarchitectural feature called simultaneous multi-threading which utilizes processor resources that would otherwise remain idle because of data dependencies within a single thread. The technique requires the insertion of a specialized delay buffer into the microarchitecture to enable comparisons between the execution result streams of the two threads. The authors presents simulation results that show the time penalty of executing two copies is 10–30% larger.

A $\mu P$ dynamic verification CSM method for detecting transient and permanent faults is proposed in [14]. A DIVA checker is proposed that recomputes the functional unit result using the instruction input operands and compares the results before allowing the instruction to commit. Although the checker design is simplified because it can leverage the processor pipeline decisions, the checker pipeline introduces area overhead for additional register file and cache ports, and for redundant functional units. Area estimates are provided in follow-on work [15] which predicts a value of $\approx 5\%$, i.e., 10 mm$^2$ in an Alpha $\mu P$ with area 205 mm$^2$.

A hardware-software high-level CSM fault detection technique is proposed in [6] that monitors software execution for anomalous behavior. Fault detection is performed at a high level by observing hardware traps and microprocessor performance counters. Although the technique is able to detect 95% of the unmasked faults, the latency for detection can be high. Although most were detected in less than 100K instructions, others take longer, up to 10 million instructions. The area and performance overheads are $\approx 0\%$ because the technique leverages only existing architectural features that are accessible by software running on the $\mu P$. However, given the large latencies in some cases, it is unclear whether this technique will be able to detect leakage sensitive faults before leakage occurs.

The authors of [7] propose a periodic built-in self-test (PBST) fault detection and diagnosis technique that utilizes a set of special instructions, called access-control extension (ACE) instructions, to access state and control $\mu P$ execution. The extended instruction set leverages the existing scan-chain intrastructure to enable access to all microarchitectural state components which keeps hardware overhead low. The technique periodically suspends execution and runs a set of manufacturing tests that are crafted to provide high fault coverage. An area overhead of 5.8% is reported for Sun's Niagara OpenSPARC T1 $\mu P$. The performance overhead depends on the length of the checkpoint interval and fault model. The minimum average performance overhead is reported as 5.5% for stuck-at fault-model testing and for a checkpoint interval of 100M instructions.

## 3. System Overview

The details of the fault campaign, fault emulation engine, fault injection and counter circuit design, FPGA testing process, computer-aided-design (CAD) tool flow, fault injection

experiment run times and fault analysis are presented in this section. As indicated earlier, the soft core implementations of the *µPs* on the FPGA are constructed using application-specific integrated circuit (ASIC) tools as a means of ensuring the emulated fault behaviors are consistent with those that would be observed in custom layout of the *µPs* .

### 3.1. Fault Campaign Characteristics

The term fault campaign is used in reference to the fault emulation system and its capabilities [16], and is defined by several attributes including the computing and communication mechanism between the fault injection manager (FIM) and fault emulation (FE) engine, the design-under-test, the fault model(s) and fault analysis techniques. A summary of the fault campaign utilized in this research is given as follows:

- The RISC-V River *µP* (Rocket) [11,17] is used as the processor-under-test, configured with a 32 KB ROM for application code and a 512 KB BRAM for scratch memory. The netlist for Rocket is generated using an ASIC synthesis and place&route computer-aided design (CAD) tool flow. The netlist is instrumented with scan-chains that connect to a set of fault injection circuits and a set of counters. The instrumented netlist is used as input to a FPGA CAD tool flow which produces the programming bitstreams for the FPGA.
- The FIM is implemented as a C program that runs on an embedded processor within a multiprocessor system-on-chip (MPSoC) FPGA. Similar to the FI architecture proposed in [18], we leverage two 32-bit high-speed, memory-mapped general purpose input/output (GPIO) registers between the processor and programmable logic (PL) components for fault injection, control and counter data retrieval.
- A set of single-bit faults emulating stuck at 0 (SA0), stuck at 1 (SA1), invert and delay fault types are inserted and reported on.
- The FE engine is constructed as a set of PL-side state machines tightly integrated with the processor-under-test. The FIM communicates with the FE engine and drives the scan chains using the GPIO registers.
- The FPGA is reprogrammed with the bitstream before each FI experiment to eliminate error propagation between FI experiments that affect non-resettable FFs and block RAMs.
- The fault emulation process includes a fault free emulation to determine the impact of each fault on the toggle (transition) activity associated with each of Rocket's circuit nodes.
- The analysis of the counter values is done off-line using the counter values collected from the fault-free experiment and each of the FI experiments.

### 3.2. System Architecture

A block diagram of the experimental design is shown in Figure 1. The Zynq UltraScale+ MPSoC incorporated on the Xilinx ZCU102 development board includes a processor side (PS) and programmable logic side (PL) side. The Linux operating system (OS) boots from a 16 GB SD card and runs on an ARM Cortex-A53 processor [19]. An ethernet channel is set up to enable communications between the host and the FIM, which is implemented as a C program on the FPGA.

The Xilinx Vivado block diagram tool is used to add an AXI-lite GPIO port to the PL-side FE engine. The block diagram in Figure 2 illustrates the connectivity to and from the two GPIO registers on the PL-side. The two 32-bit *GPIO Ins* and *GPIO Outs* registers can be directly accessed from the FIM as variables within a C program because they are memory-mapped into the PS-side address space.

As indicated earlier, the FIM dynamically reconfigures the PL side before each FI experiment as a means of ensuring the emulated Rocket processor state is reset and isolated from fault effects from previous FI experiments. Dynamic reconfiguration (DR), in contrast to dynamic partial reconfiguration (DPR), reprograms the entire PL-side, resetting all FF and Block RAM (BRAM) resources to their initialized, fault-free state. The processor

configuration access port (PCAP) shown in Figure 1 is used to accelerate the DR process, which takes approximately 215 milliseconds.

The FIM C program communicates with the FE master state machine (*Mst*) through the GPIO registers as shown in Figure 2. The FIM-controlled scan clock (*scanClk*) and scan enable (*scanen*) signals connect to clock trees on the PL-side to enable scan operations, i.e., fault injection and counter value read-out, to proceed at high speed.

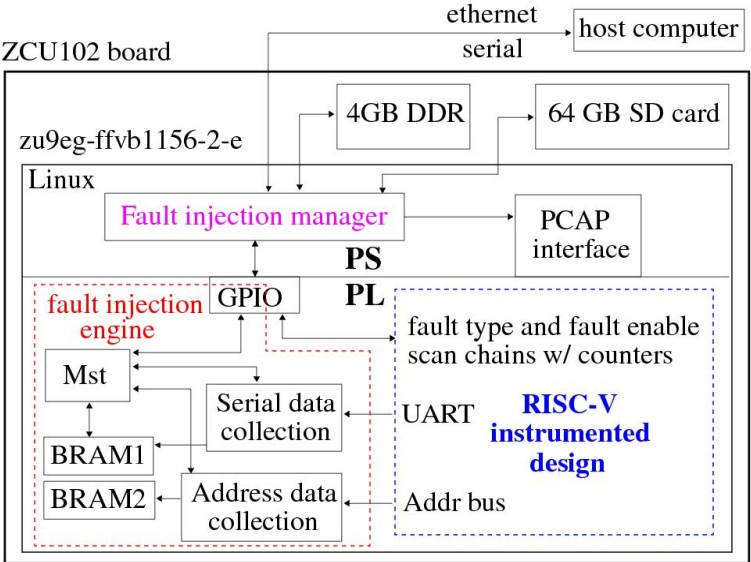

**Figure 1.** Block diagram of the experimental setup with fault injection and counter circuit scan chains for accelerating data collection.

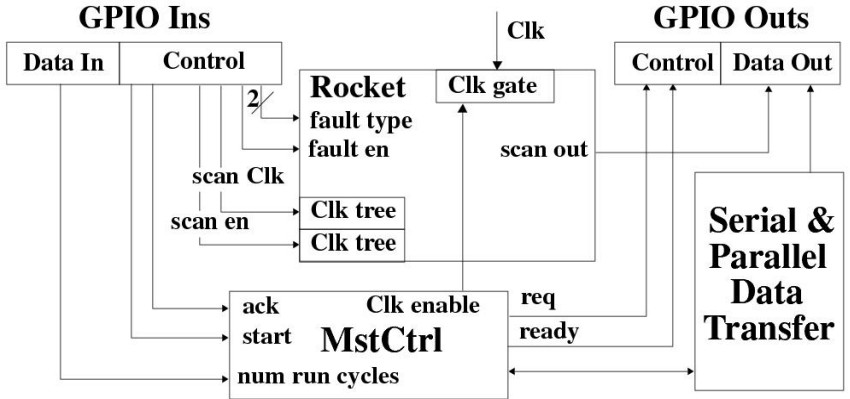

**Figure 2.** PS-to-PL-side GPIO registers for FIM and FE engine communication and scan chain implementation characteristics.

### 3.3. Fault Injection and Counter Circuit Design

The Rocket netlist obtained from the CAD tool synthesis flow contains 85,714 nodes. The netlist is post-processed to create an instrumented design with fault injection circuits inserted in series at each of the circuit nodes. A subset of the fault injection sites are also instrumented with counter circuits. The Rocket netlist and fault injection circuits use significant FPGA resources, limiting the number of counters that could be inserted to 2001 per bitstream. In order to obtain complete coverage, a set of forty-three FPGA bitstreams are created, each with 2001 counter circuits instantiated at unique fault injection sites (the last bitstream instantiates only 1714 counters). The last counter instance in each bitstream is repeated as the first counter instance in the next bitstream to enable consistency checks to be carried out on data read out from consecutive bitstream experiments. Therefore, the

FI experiments are repeated fourty-three times as a means of collecting the counter values for the entire set of 85,714 nodes.

The fault injection circuit is shown in schematic form along the bottom of Figure 3 while the counter circuit is shown above it enclosed within a red rectangle. A set of three scan chains, labeled *scan in[2:0]*, are used to control the insertion of faults. The signal *scan in[0]*, a.k.a. *fault en*, controls the insertion of the fault between the Rocket circuit node labeled *in* and *out*. The *scan in[2:1]* signal pair selects the *fault type*, which can be any one of four possible fault conditions, *stuck-0*, *stuck-1*, *delay* and *invert*.

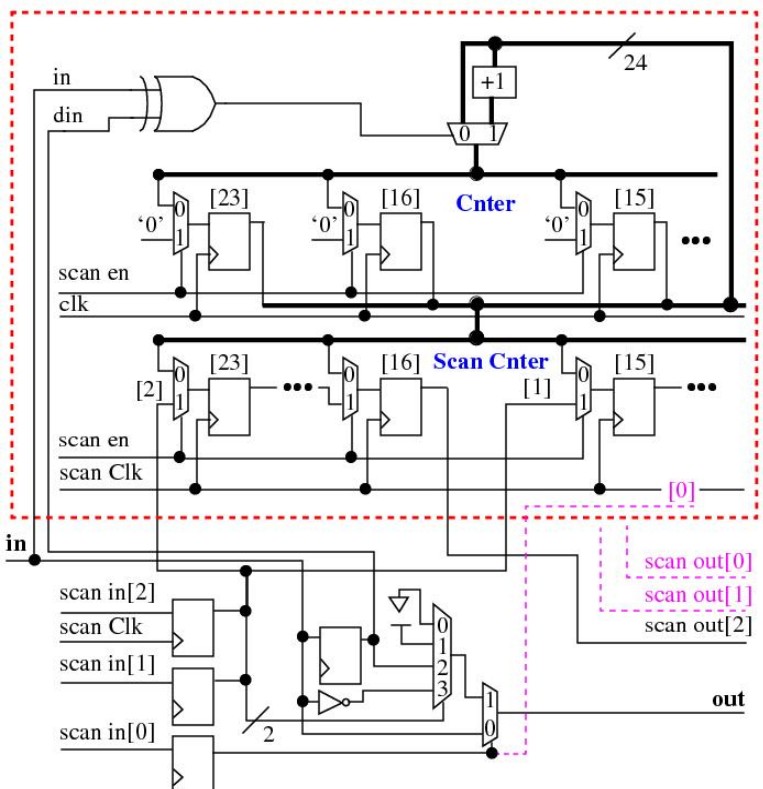

**Figure 3.** Schematic of the counter circuit integrated into the fault injection scan chain.

The delay fault component utilizes a flip-flop (FF) to insert a one-cycle delay, as a model for a fault condition that delays the signal propagation of a Rocket node to a downstream gate input. The output of this FF is also used to trigger the counter circuit, labeled *Cnter*, to increment the count when a value change (transition) occurs on the corresponding Rocket node, which is accomplished using the XOR gate logic shown along the top of Figure 3. On any particular clock cycle, if the old state of the node is a '0' and the new state is a '1', the output of the XOR gate is '1', and vise versa. The XOR gate drives the select input to a 2-to-1 MUX that selects the '+1' feedback connection from the counter when a transition occurs on the corresponding Rocket node.

During system operation, the count value in the *Cnter* is transferred to a shadow copy, labeled *Scan Cnter* in Figure 3, after the test completes. The 24-bits of the *Scan Cnter* are stitched into the scan chain by partitioning the counter bits into three 8-bit groups, one group for each of the three scan chains.

### 3.4. Testing Process

The following sequence of operations is carried out during each FI experiment to measure and then scan out the contents of the *Scan Cnters*:

1.  The PL side is reprogrammed using DR. The initial state of the FE engine asserts the *reset* signal to Rocket and disables the *Clk*, which prevents any switching activity within Rocket.
2.  The FIM configures a register within the FE engine that controls the run time of Rocket, which is specified as a terminating clock cycle count.
3.  The FIM inserts a fault using the scan inputs, i.e., *scan in* and *scan Clk*.
4.  The *scan en* is asserted and Rocket's clock is pulsed for one cycle, which clears the *Cnter* values.
5.  The *scan en* signal is de-asserted and the FIM starts the FE engine, which simultaneously releases *reset* on Rocket and enables the *Clk* input.
6.  The FE engine counts precisely up to the terminating clock cycle count and then simultaneously re-asserts *reset* and disables *Clk*. It then informs the FIM that the test has completed.
7.  The FIM pulses *scan Clk* one time with *scan en* low to transfer the contents from the *Cnters* to the *Scan Cnters*.
8.  The FIM asserts *scan en* and scans out the contents of the scan chains and reconstructs the counter values.

*3.5. Rocket Synthesis*

The goal of the synthesis and place&route CAD tool flow is to produce an implementation of Rocket that can be fabricated as a stand-alone device, i.e., an ASIC. The structural netlist produced by the ASIC tool flow is used as input to a second FPGA CAD tool flow to generate a design that can be used in the FPGA emulation experiments. The FI insertion technique ensures the FPGA version is structurally identical to the ASIC version, as a means of making the reported results meaningful to an actual application that would utilize the ASIC device. The following process is followed to meet this goal.

First, a behavioral description of Rocket, derived from Chisel [8], is used as input to an ASIC standard cell synthesis and place&route CAD tool flow using the ASAP7 7 nm FinFET standard cell library [20]. The Synopsys Design Compiler [21] is used to generate a gate-level netlist and Cadence Encounter [22] is used to perform place&route to generate a layout representation. A netlist is extracted from the layout representation, which is then converted into an instrumented design with scan-controlled FI circuits and counters using a custom C program. As mentioned earlier, a set of 43 instrumented netlists are created with the C program, each with a distinct set of counters.

Xilinx Vivado [23] is used to generate the bitstreams using the instrumented netlists as input. The FI circuits and scan chains prevent Vivado from optimizing the original ASIC netlist structure. The FPGA resources utilized to implement the emulated design are given as follows:

Look-up Tables (LUTs): 153520 : 56.01%
LUT RAMs:              314     : 0.22%
Flip-Flops:            403643 : 73.64%
BRAM:                  166     : 18.20%

Vivado takes approximately 6 h to synthesize, implement and generate a bitstream for each netlist. Although this process takes about a day to complete with several servers running multiple jobs in parallel, the run time for the data collection is much larger, and therefore, bitstream preparation represents only a small fraction of the total FI campaign run time.

*3.6. Counter Location*

The counter locations within the major functional units of Rocket are classified according to the names in the synthesized netlist. The counter plots presented in the following place counters within the same functional together as a group to assist with illustrating the impact of faults at the functional unit level. Table 1 gives information regarding the

functional unit names listed across the top of the figures. Note that this is a very broad characterization of Rocket's internal functions as they relate to Rocket's pipeline shown in Figure 4.

**Table 1.** Counter number to functional unit mapping.

| Functional Unit | Counter Range | Description |
|---|---|---|
| Txxx | 0 thr 29,283 | Miscellaneous logic |
| alu | 29,284 thr 34,894 | Arithmetic logic unit, adder, etc |
| csr | 34,895 thr 60,937 | Control and status registers |
| div | 60,938 thr 76,502 | Divider |
| ex_ctrl | 76,503 thr 80,510 | Ctrl logic execute stage |
| io & mem | 80,510 thr 85,713 | I/O and memory logic |

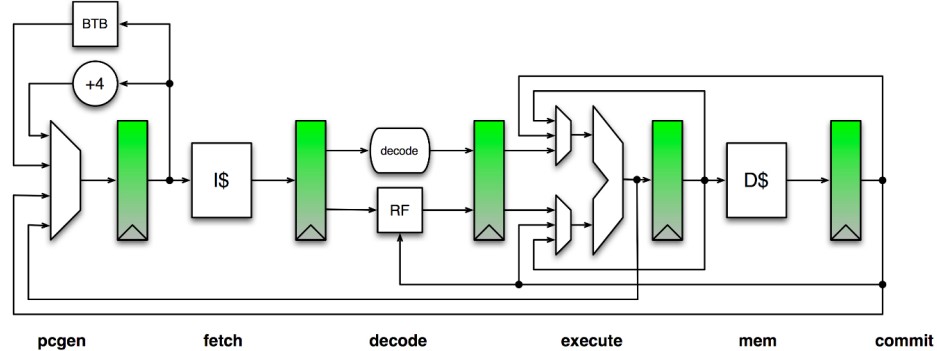

**Figure 4.** Rocket core pipeline [24].

### 3.7. FI Experiment Run Times

The FI experiments consist of emulating Rocket running one encryption operation under fault-free and fault conditions, each at a series of stop points, as a means of evaluating fault detection latency. A C program running on one of the Cortex A53 APUs serves as the FIM, which automates and coordinates the entire testing process. In the fault experiments, an outer loop in the FIM configures each of the faults into the scan chain (see Figure 2), one at a time, while the inner loop configures the FE engine with a data word that controls the run time, given as the number of clock cycles.

An initial set of experiments are run in which Rocket fully executes the AES encryption algorithm as a means of determining which of the 85,714 faults of each fault type are masked (benign) or unmasked (active). The DR operation takes approximately 215 milliseconds (ms), while a full execution run allows Rocket to run for $2^{22}$ clock cycles, which takes approximately 262 ms. The scan operations need to scan 85,714 fault cells and $8 \times 2001$ counter cells, which takes approximately 49.6 ms, computed as 488 nanoseconds/scan operation $\times (85,714 + 8 \times 2001)$. Other C program execution time within the FIM takes an additional 73 ms. Therefore one full execution takes approximately 600 milliseconds. With 4 fault types, total run time for these experiments running on a single copy of the ZCU102 is $4 \times 85,714 \times 0.6 \approx 57$ h per bitstream. With 43 bitstreams, a complete data set was collected in approximately 51 days running on 2 copies of the ZCU102.

The 85,714 counter values collected from the FI experiments are compared with fault-free values to determine which faults are masked (the process of distinguishing masked faults is described in the following section). After eliminating the masked faults, the total number of active (unmasked) faults reduces from $4 \times 85,714$ faults = 342,856 to 101,729 faults, more than a 3X reduction.

Using only the active faults, we identified the top five most sensitive counters (discussed later) and focused subsequent experiments only on them. In a second set of experi-

ments, the latency for each of the active faults is determined using a binary search process. Although the details are presented later, each search requires 15 iterations to identify the point at which an active fault introduces an anomaly in the execution behavior of Rocket at a resolution of 128 clock cycles. The total run time for these experiments is given by $101,729 \times 15 \times 0.6 = 10.6$ days per counter on a single ZCU102.

A third set of experiments were run that possessed these same run time characteristics but using only the set of 340 faults that leak key and plaintext information, which we refer to as severe faults here and in previous work [9]. The smaller fault set size reduced run times to one hour for each fault. The total run time of all experiments using 2 copies of the ZCU102 is 51 days + 25.3 days + 7 days $\approx$ 84 days.

*3.8. Active and Benign Faults*

The counters provide high resolution observation over the impact of the faults across the entire Rocket internal architecture. The fault detection capabilities of the counters is herein referred to as fault coverage, a metric similar to the same term used within the manufacturing test community. The counters provide observation of internal node behavior, in contrast to the more limited observability typically available at the primary outputs (or scan FF inputs) in microprocessors instrumented with design-for-testability features.

Although our analysis is focused on severe faults, we also want to report on the collateral coverage provided by the selected counters for all of the faults. However, a large fraction of the 342,856 faults are masked, and hence are benign, i.e., they do not introduce anomalous behavior in the execution of the AES algorithm, and should be excluded from the analysis. Active or unmasked faults, on the other hand, are defined as faults that introduce significant changes, i.e., cause Rocket to lock-up, or result in corrupted ciphertext output. Active and benign faults can be distinguished by comparing the counter values from the fault-free emulation with those obtained when the fault is enabled. For simple cases in which all 85,714 counters store the same counts in both emulations, the fault can be classified as benign or masked.

A second simple benign case which can be easily identified involves faults that impact only the counter value at the location of the fault site. For example, activating a stuck at 1 fault on some node, where the fault-free value of the node remains at 0 for the entire execution, will cause the counter to increment by 1 at that location in the FI experiment. This fault can also be classified as benign under the condition that the remaining 85,713 counters match the fault-free counts. The total number of faults classified as benign under these conditions is given as follows for each fault class:

Stuck-at-0: Masked: 43,868
Stuck-at-1: Masked: 40,160
Delay:      Masked: 61,046
Invert:     Masked: 22,69

The strict conditions associated with the criteria described above represent a conservative measure of masked faults. A second source of information that can be used to help identify masked faults is to compare the serial output and address bus behavior as Rocket executes with that produced from the fault-free emulation, identical to the approach taken in previous work [9]). The number of faults classified as benign under these conditions are given as follows:

Stuck-at-0: Masked: 64,540
Stuck-at-1: Masked: 63,230
Delay:      Masked: 71,407
Invert:     Masked: 56,295

From these two sets of results, it is clear that there are a large number of faults that are neither active or benign, and instead fall into a 'indeterminate' category as it relates to classifying Rocket execution behavior. For example, there are 64,540 − 43,868 = 20,672

indeterminate Stuck-at-0 faults. It is possible to partition these indeterminate faults into masked and active by relaxing the strict constraints described above, and additionally classify faults as benign if more than one counter value in the FI experiment is different than the fault-free values but the remaining counters match the fault-free values and the AES serial output and address behavior remain identical to the fault-free emulation values. The relaxed constraints handle a common scenario in which the output of the activated fault drives combinational logic gate inputs downstream that cause the associated counters to increment by 1, but are blocked by internal state from further interactions with Rocket's execution. The following table gives the final number of faults classified as active and benign in our experiments:

Stuck-at-0: Active: 24,655 Masked: 61,059
Stuck-at-1: Active: 26,132 Masked: 59,582
Delay:      Active: 16,546 Masked: 69,168
Invert:     Active: 34,396 Masked: 51,318

We recognize that many of the faults classified as benign in these sets would in fact introduce anomalies in Rocket given a different work load, e.g., program to execute. However, our goal here is to determine the number of required counters needed to detect all severe faults and to evaluate the collateral coverage of the active faults using a frequently used program that can serve as an estimate of the overhead and effectiveness for this type of countermeasure approach. As we discuss later, future work will focus on using automatic test pattern generation (ATPG) and an engineered executable to minimize the number of required counters while maximizing the coverage.

### 3.9. Severe Faults

In previous work, we identified a total of 340 severe faults, partitioned as follows based on the fault type introduced [9]. Note that the severe faults are by definition also classified as active, and are included in the active subset.

Severe Stuck-at-0 (SA0): 76
Severe Stuck-at-1 (SA1): 49
Severe Delay:            129
Severe Invert:           86

Delay faults represent the largest class of severe fault types, and as discussed earlier, are also more likely to occur before hard faults over normal wear out periods of operation.

### 4. Experimental Results

The fault detection and latency results for the counter-based CM are presented in the following sub-sections. An analysis that includes all 101,729 active faults, referred to as the All-faults class, is presented in Section 4.1. A separate analysis of the 340 severe faults, referred to as the Severe-faults class, is presented in Section 4.2. The latency analysis of the two fault classes as well as the latency associated with key and/or plaintext leakage on the serial port is presented in Section 4.3. A summary of the results obtained for the Potato $\mu P$ are given in Section 4.4, followed by a discussion of overhead and on-going work in Sections 4.5 and 4.6.

### 4.1. All-Faults Analysis

As discussed in Section 3.8, the data collected for all 85,714 counters in the initial set of experiments are used to classify the 85,714 faults as active and benign. In this section, a small subset of five counters are identified as providing the highest coverage of all active faults, which are later compared with the results obtained for severe faults.

An important first result is the number of active faults that each counter is able to detect. A fault is counted as detected if one or more counter values under the FI experiment with the fault enabled is different than the value measured under the fault-free FI experiment.

The results are presented as color-coded points in Figure 5, with the counters partitioned along the x-axis according to the functional unit in which they are placed. The total number of active faults detected by the most sensitive counter for each fault type, as well as the fraction detected by the top three most sensitive counters is given in the following.

Top counter: Stuck-at-0: 22,938 Top 3 counters: 94.14%
Top counter: Stuck-at-1: 23,975 Top 3 counters: 93.07%
Top counter: Delay:      15,131 Top 3 counters: 93.88%
Top counter: Invert:      31,623 Top 3 counters: 93.43%

These results indicate that a large fraction of the active faults in each fault class are detected by small number of counters. More importantly, the top three counters are the same for stuck at 0 (SA0), stuck at 1 (SA1) and invert analyses. The top three most sensitive counters for the delay faults are distinct but the top three counters for SA0, SA1 and invert are in the top six most sensitive counters for delay. This suggests a stronger result that only a set of six counters are needed to detect a large fraction of all the faults. Lastly, this same set of counters also detects all severe faults.

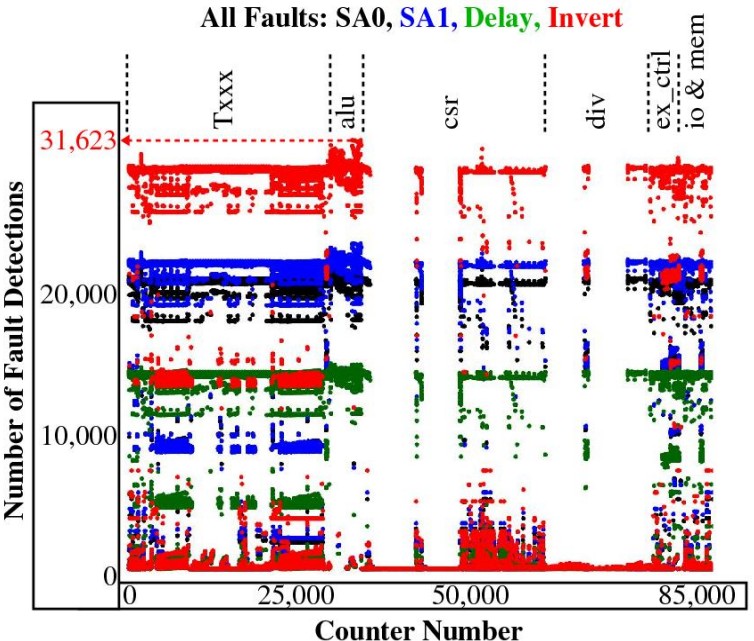

**Figure 5.** The number of faults detected by each counter. The counter number is plotted along the x-axis against the number of faults detected along the y-axis. The points are color-coded according to the fault type, with black, blue, green and red showing SA0, SA1, delay and invert fault detections, resp.

The counters associated with these maximum detection percentages are located in the ALU and register file categories, and in particular, on different bits within the execution control unit to the by-pass MUX of the register file, as given by Table 2. The faults detected by these counters overlap significantly (as implied by the detection percentages presented above) and therefore, any one of them is sufficient to serve as a monitor for detecting the faults with only a small decrease in fault coverage (later we show that each counter fails to detect approx. 9% of the faults).

**Table 2.** Top 5 All-Fault Counter Mapping.

| Counter ID | Counter Number | Location |
|------------|----------------|----------|
| $AFC_0$ | 32,968 | Ex Ctrl to By-pass MUX of Reg. File |
| $AFC_1$ | 33,997 | Ex Ctrl to By-pass MUX of Reg. File |
| $AFC_2$ | 33,442 | Ex Ctrl to By-pass MUX of Reg. File |
| $AFC_3$ | 29,596 | Ex Ctrl to By-pass MUX of Reg. File |
| $AFC_4$ | 29,600 | Ex Ctrl to By-pass MUX of Reg. File |
| $AFC_5$ | 33,213 | Ex Ctrl to By-pass MUX of Reg. File |

Figure 6 plots the cumulative detections for a subset of the counters which are needed to detect all active faults. The counters are sorted by the maximum number of faults that each detects, i.e., most sensitive to least sensitive. The number of counters needed to detect all of the faults in the All-faults class is given as 167, 188, 121, 172, but as indicated above, most of the faults are detected by the first three counters.

The presence of the fault changes the counts by different amounts depending on the location of the counter. Faults that occur within frequently used functional units impact counts more dramatically. Figure 7 plots the sum of the counter differences for counters with counts larger than those measured in the fault-free emulations across all active faults. The large positive sums associated with the Txxx, alu, ex_ctrl and io & mem regions indicate that significant switching activity occurs in these counter regions and the faults that occur there dramatically impact the switching activity. Although not shown, similar behavior is observed in the counter sums associated with nodes with counts smaller than those measured in the fault-free emulation.

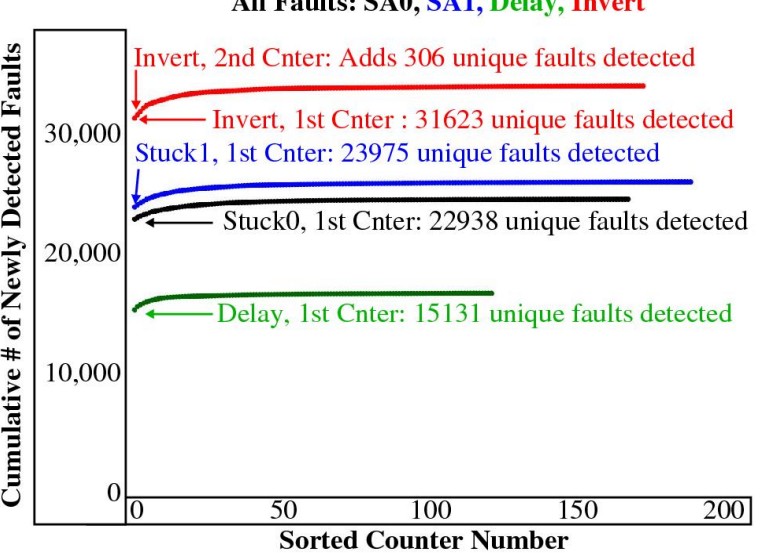

**Figure 6.** The cumulative number of newly detected faults detected by each of the counters.

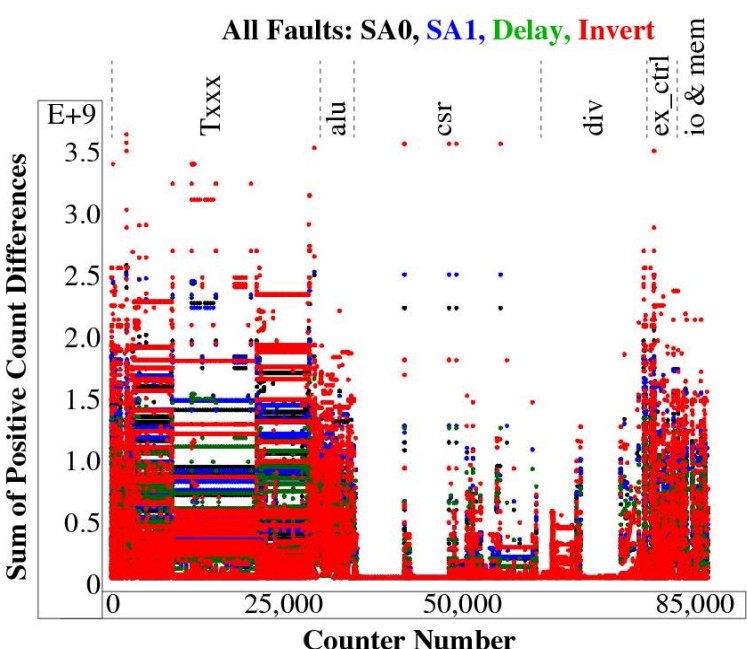

**Figure 7.** Sum of the positive counter differences with each fault enabled, one-at-a-time, for each counter (x-axis), indicating regions with additional switching activity when the faults are activated. The fault-free counts are subtracted from the counts obtained in each FI experiment.

### 4.2. Severe Faults

The number of faults detected by the counters using only faults from the Severe-fault class is shown in Figure 8. The numbers given along the right side of the figure list the maximum number of severe faults of each type. The results illustrate first that delay faults dominate the leakage faults in contrast to the results from the all-fault analysis which show invert faults as the dominant fault class. Second, there are a large number of counters that detect all of the severe faults of each type. The implication here is that a single counter is sufficient as a countermeasure to detect all severe faults and that there exists a wide range of counters (and regions) that can be selected for the location of the monitor. Equally important is the illustration that a large fraction of the counters that detect all faults of each type overlap with each other. In total, 28,767 counters detect all 340 severe faults.

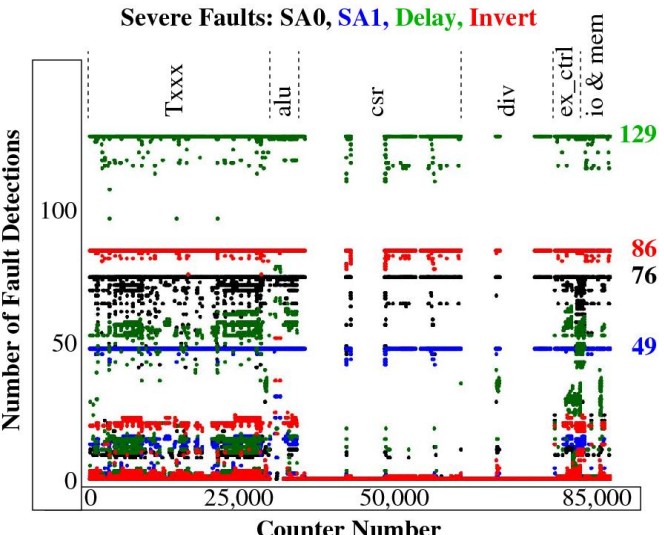

**Figure 8.** The number of severe faults detected by each counter. Annotations are identical to those given in Figure 5. The total number of severe faults per fault class is given along the right side.

*4.3. Latency Analysis*

In this section, we analyze the latency of the counters in detecting the 340 faults in the Severe-faults class, and present an analysis showing the trade-off between the number of counters and the worst case latency for detecting these faults. Latency refers to the number of clock cycles that are required before the fault creates an anomaly in the internal state of Rocket and changes the counter value from that measured under fault-free conditions. We also carry out a latency analysis for the severe faults using the top six counters identified for the All-faults class to determine if targeting severe faults reduces latency and improves the results.

The process used to determine fault detection latency involves first measuring the fault-free counts at a series of incrementally increasing run times of Rocket, referred to as stop points. The run times are expressed in clock cycles. Once the fault-free counts are available, the FIM reads the fault-free data file and carries out a binary search to find the minimum latency of each fault. Two counter groups are considered. The first set is the group of six counters identified in the All-faults analysis. The second set is the group of 28,767 counters identified earlier as detecting all 340 severe faults.

A linear sweep algorithm is used to record the toggle counts for each of the counters at a series of stop points in a fault-free emulation of Rocket, with the stop points defined as incremental multiples of 128 clock cycles. The number of clock cycles required in the fault-free emulation of the AES algorithm is upper bounded at 1,705,344 clock cycles. However, a subset of the faults increase the execution time of the algorithm. In order to capture the results of these run-away executions, we run Rocket for $2^{22}$ or 4,194,304 clock cycles, and collect toggle counts at each of the 32,768 stop points that occur over this longer interval. The latency for each of the faults is found using the fault-free data and a binary search process, that proceeds as follows for each counter and for each fault.

- The PL side is dynamically reprogrammed (DR) with the FI bitstream to ensure fault effects do not propagate across re-executions of Rocket.
- The fault is inserted using the scan chain.
- Rocket is run and then halted after 2,097,152 clock cycles, exactly half the maximum number.
- The count value stored in the target counter is compared with the fault-free value. If the count value does not equal the fault free value, the search continues using a stop point equal to 1,048,576 clock cycles (1/2 the previous value). If it is equal, the search continues using 3,145,728 clock cycles as the stop point (1.5X the previous value).
- Continue the search until two consecutive stop points are found, one in which the counter value matches the fault-free value and the next where a mismatch occurs.

Each search takes 15 iterations to find the point at which the fault effect creates the anomaly, within the tolerance window of 128 cycles. Each iteration takes ≈600 milliseconds, yielding a search time of ≈9 s per fault. Note that it is possible to reduce the run time to ≈5.8 s per fault by eliminating DR but unfortunately, asserting reset between runs is not sufficient to restore the fault-free state of Rocket for a large fraction of the faults.

4.3.1. Latency Analysis of the All-Faults Class

The curves in Figure 9 show the count behavior as a function of the number of clock cycles plotted along the x-axis for the six counters identified earlier as the most sensitive in Table 2. As noted, there are 32,768 data points associated with each curve. Except for two brief stall points, the counter values continuously increase as the number of run cycles increases. This characteristic is somewhat intuitive for counters that are the best candidates for monitoring and detecting fault effects. The count values for counters $AFC_1$ and $AFC_0$ are nearly identical. This occurs because the counters are located in the same logic cone of the design. Therefore, counter $AFC_0$ is not considered further in our analysis. It is noteworthy that the second stall point is in the region associated with the completion of the AES encryption algorithm.

**Fault-Free Counter Analysis**

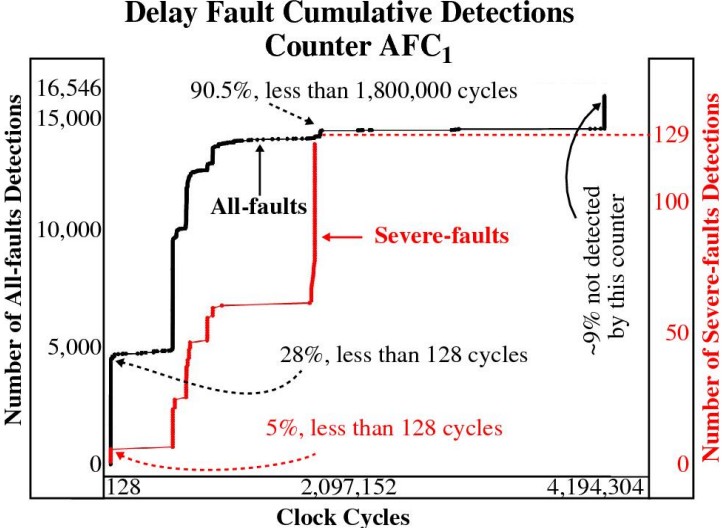

**Figure 9.** Fault-free counts as a function of stop point for the six most sensitive counters from the All-faults analysis.

The latency results for $AFC_1$ are plotted in Figure 10 showing the cumulative fault coverage of All-faults in black and Severe-faults in red as the number of clock cycles increases along the x-axis. The jump to nearly 5000 by the black curve on the left side of the graph indicates that ≈28% of the faults from the All-faults class are detected very early in the run of the algorithm, i.e., in less than 128 clock cycles. Approximately 90% of them are detected within the execution time of the AES algorithm (≈1.7 million cycles) and ≈9% are not detected at all by this counter. The cumulative coverage for the 129 severe delay faults plotted in red shows ≈5% are detected in less than 128 clock cycles while detection of the full set takes ≈1.7 million clock cycles.

**Delay Fault Cumulative Detections
Counter $AFC_1$**

**Figure 10.** Delay fault cumulative fault detection results for Cnter $AFC_1$ with run cycles plotted along the x-axis and number of fault detections along the y-axis.

The cumulative delay fault coverages associated with all five All-faults counters from Table 2 are shown in Figure 11. The behaviors are strikingly similar for both the All-faults and Severe-faults classes, despite the fact that the increasing count rates shown in Figure 9 differ. A key consideration here is determining whether the severe faults detected in the region of 1.7 million clock cycle are detected before the serial output begins to leak

information, and whether there exist counters that provide lower latencies in the set of 28,724 counters that detect all severe faults.

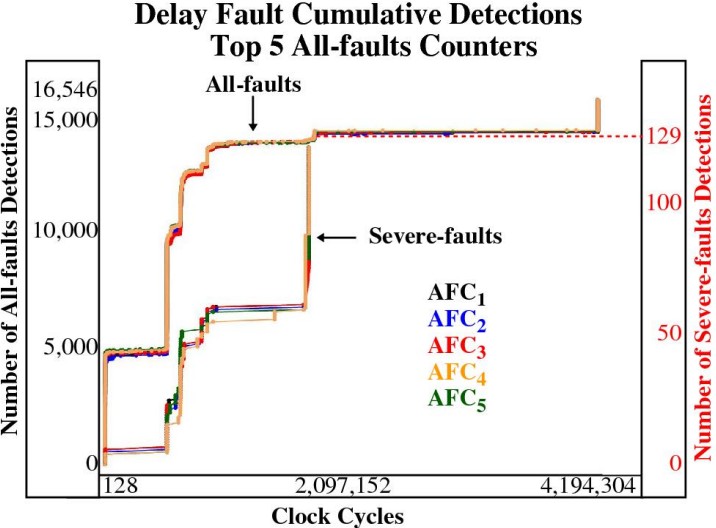

**Figure 11.** Delay fault cumulative fault detection results for all five of the most sensitive counters with run cycles plotted along the x-axis and number of fault detections along the y-axis.

4.3.2. Latency Analysis of the Severe-Faults Class

The goal of the All-faults analysis is to select a small set of counters that detect the largest possible fraction of all active faults. Therefore, the counters selected may not be optimal for minimizing the detection latency of the severe faults. In this section, we focus on identifying the set of counters that provide the smallest possible severe fault latency, and compare the results with the top five All-fault counter latencies. We conclude that although there exists a set of severe fault counters that can reduce latency significantly, the number required is large and impractical. Moreover, down-selecting to a small set of the best Severe-fault counters provides no significant benefit in latency over any of the top five counters identified from the All-fault analysis. Lastly, the top five All-fault counters detect all severe faults before any leakage occurs on the serial port output, and therefore, any one of them meets the goals of our proposed reliability monitoring scheme.

As indicated earlier, a total of 28,767 counters detect all severe faults. We carried out a set of fault emulation experiments using this large group of counters to determine which subset of counters provides the smallest possible latency for each of the 340 severe faults. Note that the binary search process allows the latency of only one counter to be determined in each fault emulation experiment, in particular, we always select the counter that provides the minimal latency for the emulated fault. Given the counters are partitioned across a set of 43 different bitstreams, the results of this analysis identified a set of 1840 unique counters, each of which identified 1 of 340 faults at minimum latency. Further processing reduced this set to 168 counters, each of which provided the lowest overall latency across all bitstream results.

The smallest latencies associated with this set of counters is shown in Figure 12, with the 340 severe faults numbered along the x-axis against their minimum latencies along the y-axis. These results represent the best case for the proposed counter-based detection method. The worst case latency is 667,136 clock cycles, which is associated with a delay fault. Given the total number of clock cycles for the AES encryption operation is ≈1.7 million, these best case latencies indicate that all severe faults are detected in less than 40% of the total execution time. However, as we will show, some severe faults start leaking information on the serial port early (well before encryption finishes), and therefore, deciding whether leakage occurs needs to be determined on a fault-by-fault basis.

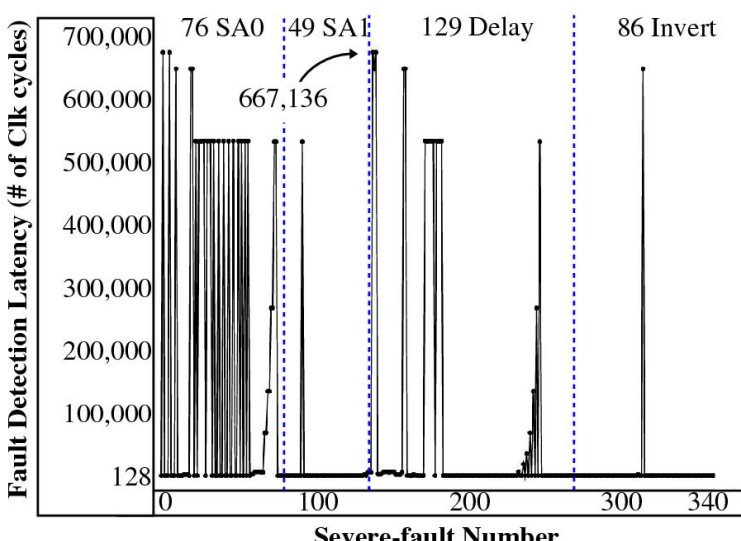

**Figure 12.** Best case latency in clock cycles for one of the counters in a set of 28,767 counters to detect each of the 340 severe faults. Each of the counters in this set detects all severe faults.

A second set of fault emulation experiments is then carried out using only the 168 smallest latency counters. In these experiments, each of the 340 severe faults are emulated with only a single counter considered in the binary search process, which allows the latencies for each fault and for each counter to be determined. The analysis of minimum latency using this complete data set reveals that some counters provide the same minimum latency for some faults, and therefore, only 99 counters are required to provide complete coverage at minimum latency.

The drawback of targeting the minimum latency for each of the 340 severe faults is the large number of counters required, i.e., 99 as indicated. Moreover, given that leakage always starts occurring at clock cycles well after these minimum latencies (as we show below), detecting the severe faults at minimum latency is not required. The curve shown in Figure 13 plots the overall worst case latency as the number of counters is strategically reduced, using an algorithm that removes counters that have the fewest number of minimum detection latencies, one at a time. Recall that all of the counters used in this analysis detect all severe faults. Therefore, when a minimum latency counter is removed, the faults that it detects are replaced with a counter that does not detect the fault at the minimum latency. The graph shows that the overall worst case minimum latency increases as counters are removed, up to worst case latency of ≈1.7 million clock cycles where only one counter is required to detect all severe faults.

The trade-off between the number of counters utilized and the overall minimum latency is not a smooth monotonically decreasing curve unfortunately. The precipitous drop from 55 counters to 1 indicates that a large increase is required in the number of counters to reduce overall latency from ≈1.7 million to ≈1 million clock cycles.

Detecting severe faults at 1.7 million clock cycles is close to when leakage starts to occur on the serial port so a closer inspection is required to determine if one or more of the 168 counters identified in the down-selection process detects all 340 leakage faults before any leakage occurs (we use the original, larger down-selected counter set here to be comprehensive). A similar analysis is carried out using the five counters from the All-faults analysis and the results compared, as a means of determining if targeting severe faults provides any real benefit over the counters selected in the All-faults analysis.

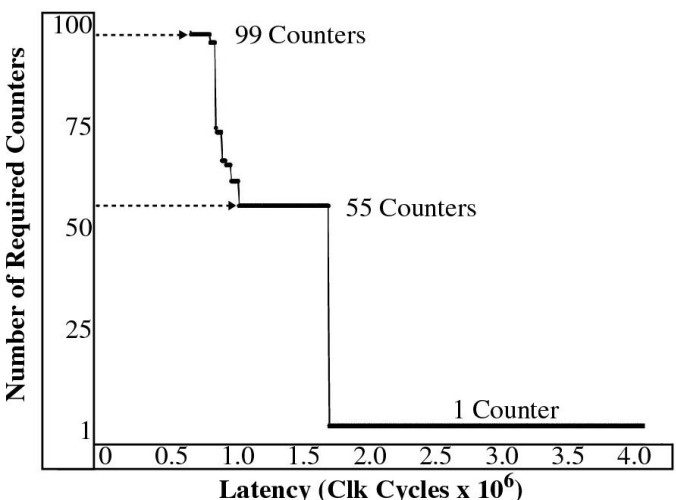

**Figure 13.** Trade-off showing the number of counters needed (y-axis) for a given minimum severe fault detection latency plotted along the x-axis.

The curves in Figure 14 show the latency results for delay faults where the latencies of the 168 counters are superimposed on the latency curves for the 5 best counters from the All-faults analysis. The best case curve is one that rises quickly and remains to the left and above the other curves, which reflects the scenario where larger numbers of severe faults are detected at smaller latencies. Although the black curves do not exhibit this ideal characteristic, they remain close to the best curves among the 168 counters that target the severe faults (one is identified as $SF_1$ in the figure). This is an important result that states that targeting severe faults in the counter analysis does not provide any significant benefit over an analysis that targets all faults. The curves in Figure 15 support this result, where the cumulative latencies for the All-faults counter $AFC_1$ are plotted against the cumulative latencies for the best of the severe fault counters, for each of the severe faults of each fault type. Again, the black curves associated with the All-faults counter are close to the best of severe faults counters (red curves), and are actually better in some clock cycle regions.

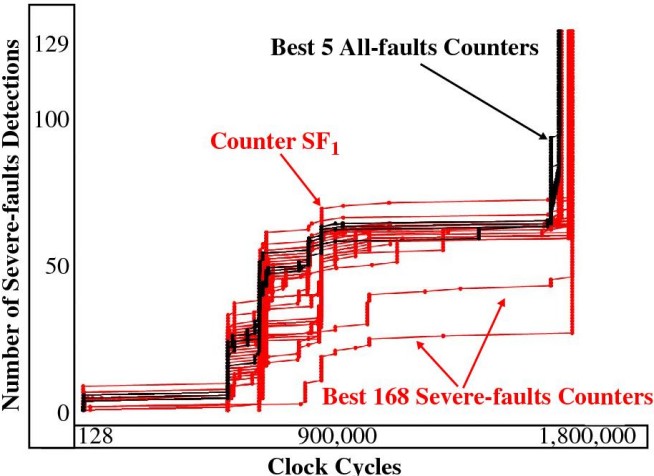

**Figure 14.** Cumulative detection latency of delay severe faults using top 5 All-faults counters and the best 168 of the Severe-faults counters.

**All Severe-faults Cumulative Detections**

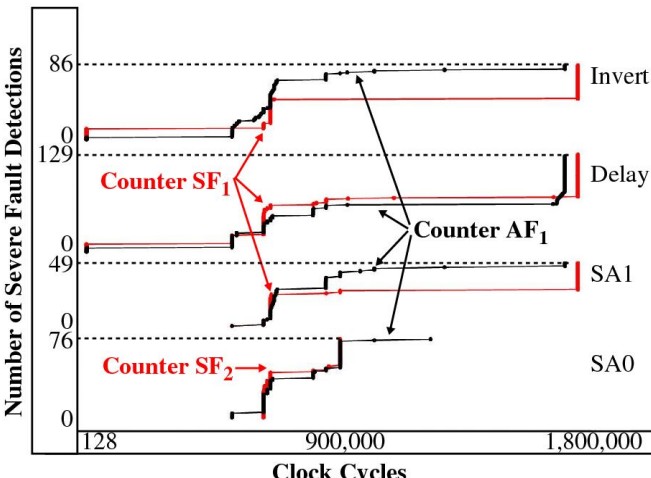

**Figure 15.** Cumulative detection latency using the best All-faults and Severe-fault counters, illustrating the best All-fault counter detects all severe faults with latencies similar to the best of the Severe-fault counters.

### 4.3.3. Latency Analysis of the Serial Port Leakage

In this last section, we compute the latencies associated with the first leakage event on the serial port, i.e., the clock cycle in steps of size 128 at which the first plaintext byte or key byte appears on the output of the serial port. The red points in Figure 16 plot the serial port latencies for each of the severe faults shown along the x-axis. Superimposed as a set of black points are the latencies for counter $AFC_1$. In all cases, the black points have smaller latencies than the red points, and therefore, the leakage event is detected before it occurs. A subset of the differences between the serial port latencies and the latencies for counter $AFC_1$ is plotted in Figure 17. Although there are some cases where the difference latencies are close to zero (the smallest value is 17,536 clock cycles) and the leakage fault is detected just-in-time, for most cases, the leakage fault is detected well in advance of the leakage event.

**Severe-faults Leakage Analysis**

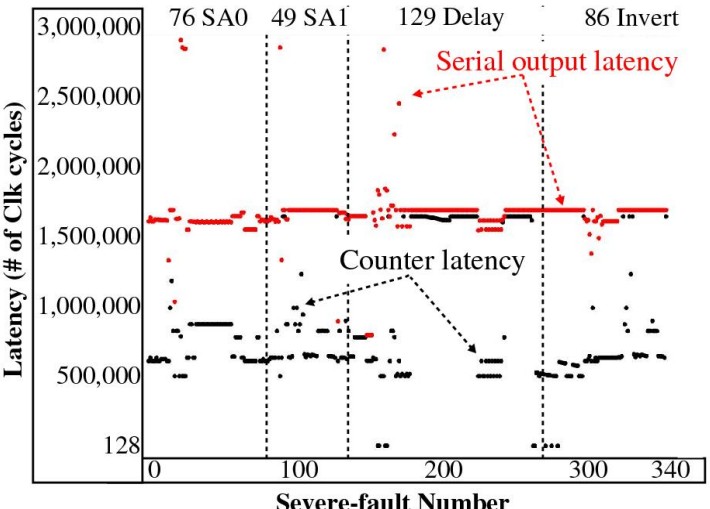

**Figure 16.** Counter $AFC_1$ latencies are plotted in black for each of the severe faults on the x-axis. The serial port output latencies are plotted in red and represent the number of clock cycles required before the first plaintext or key byte appears on the output. Although some detection latencies for the counter are close to the leakage latency on the serial port, none are larger.

**Severe-faults Leakage Analysis**

**Figure 17.** A subset of the difference latencies given as (SerialPortLatency-$AFC_1$ CnterLatency), obtained by subtracting the curves in Figure 16 and zooming in on the region close to 0.

Similar results are obtained using the remaining 4 All-faults counters. Given the All-faults counters maximize collateral coverage and are able to detect severe faults at similar latencies when compared with any of the best Severe-faults counters, we conclude that any one (or more) of the All-faults counters are the best choice(s) for the countermeasure.

The following serves as a summary of the results presented in this section:

- A set of 5 All-faults counters are identified which detect all 340 severe faults while also maintaining fault coverages in excess of ≈90% for faults in the All-faults class.
- A large set of more than 28,000 Severe-faults counters are found to detect all 340 severe faults, while a smaller subset of 99 provide the absolute minimum latencies for faults in the Severe-faults class.
- The top All-faults counter is able to detect all severe faults with latencies similar to the best of the Severe-faults counters.
- The top All-faults counter is able to detect all severe faults with latencies less than the latency of the first leakage event on the serial port and therefore, only one counter is needed to meet the goals of the proposed countermeasure.

*4.4. Potato RISC-V Results*

In this section, we summarize the results obtained from the application of the counter-based CM to a second RISC-V architecture, called Potato [10], as support for the wider applicability of the method. The Potato $\mu P$ is described directly in VHDL, i.e., is not auto-generated from Chisel as is true of Rocket, and therefore, it was possible to more precisely trace the functionality of the nodes associated with the top counters to a specific pipeline stage and functional unit within the RISC-V architecture.

Unlike Rocket, the top three most effective counters for detecting all 56,000 active faults in the All-Faults class of Potato are the same for all four fault types. Similar to Rocket, the fault coverages of the top counters for both $\mu Ps$ are ≈90%, and only one counter is needed to detect all severe faults. Moreover, the nodes associated with the top counters in both the Rocket and Potato analyses are clustered in one region of the design and both are in the fan-in cone of the forwarding logic to the register file inputs. In Potato, the nodes are located in the branch comparator module of the execute pipeline stage, and in particular, within the reduction logic network constructed by the synthesis tool to implement the comparison operators, e.g., equality, inequality, etc., for branch instructions.

The register file forwarded inputs are used as the inputs to these reduction logic operators. Moreover, the reduction operations are always performed independent of the instruction's opcode, i.e., the instruction does not need to be a branch instruction. Therefore fault effects that propagate to any bit of these register file inputs will, with high probability, manifest as a change within the reduction logic networks and on the nodes monitored by the top counters. This observation suggests that these nodes have architectural significance

from a fault propagation perspective, and explains why they are being selected as the nodes associated with the top counters.

*4.5. Overhead Comparison*

In this section, we compare the performance and area overhead of the proposed counter-based CM with previous work. Given our focus on leakage sensitive faults, the comparison with previous techniques, which target complete fault coverage, is challenging and somewhat unfair, i.e., is apples-to-oranges. Moreover, continuous symptom monitor (CSM) and periodic built-in self-test (PBST) techniques themselves represent an apple-to-oranges comparison challenge given the differences in operational characteristics and whether fault-free data is available to the methods. Table 3 summarizes the area and performance overheads associated with the most closely related techniques, with '-' indicating that no data was provided.

**Table 3.** Overhead comparison of proposed CM method with the most closely related methods.

| Author, et al. | Method | Area Overhead | Performance Overhead |
|---|---|---|---|
| Rotenberg [4] | CSM | - | 10–30% |
| Weaver [15] | CSM | 4.9% | 3.16% |
| Li [6] | CSM | $\approx$0% | $\approx$0% |
| Constantinides [7] | PBST | 5.5% | 5.8% |
| Counter (this work) | PBST | 1.5% | 1.7% |

The proposed counter-based CM has a small overhead in comparison with most of the existing methods. The counter circuit is shown within the red rectangle of Figure 3 and consists of two 24-bit counters. Area overhead is determined by processing a behavioral description of the counter through the Synopsys Design Compiler [21] using the SAP7 standard cell library [20]. The synthesis report indicates the counter can be constructed using 113 combinational logic cells and 48 FFs, and has an area of 339 $\mu$m$^2$. The results presented in this paper indicate that no more than five copies of the counter are needed, yielding an overhead of 1695 $\mu$m$^2$. In contrast, the area metrics associated with the Rocket core are 34,196 combinational logic cells and 5262 FFs, with an area of 336 $\mu$m $\times$ 334 $\mu$m $\approx$ 112,224 $\mu$m$^2$. The fractional area overhead is given as 1695 $\div$ 112,224 $\times$ 100 $\approx$ 1.5%.

The performance overhead is estimated using a checkpoint interval of 100 million instructions (similar to the ACE technique reported above). Unlike the ACE methodology, the number of scan clock cycles is very small (120 with 5 24-bit counters) and nearly all of the self-test time is attributed to program execution. Using the full run time of the AES algorithm, the performance overhead is estimated as 1.7 million $\div$ 100 million $\approx$ 1.7%. An alternative executable test program is proposed in Section 4.6 that is expected to reduce the number of self-test clock cycles significantly.

*4.6. Next Steps*

Our longer range goal is to leverage the proposed counter-based CM as a means of periodically assessing the health of a microprocessor system. The high levels of coverage obtained from experiments carried out in this paper suggest that it is possible to obtain significant fault coverage using a small set of strategically placed counters. It follows that an even better result can be obtained by 'engineering' a specialized executable, in contrast to the AES binary executed in the experiments carried out in this paper. The engineered executable could be constructed with the assistance of automatic test pattern generation (ATPG) tools. ATPG is used by the manufacturing test community to derive test vectors for detecting faults after manufacturing and can be used here for a similar purpose. The goal of ATPG is to derive a small set of test vectors that provide high levels of fault coverage.

When applied for the purpose of reliability monitoring of leakage sensitive faults, high-coverage ATPG vectors can be used to derive a set of instructions that sensitize fault effects to one or more strategically placed counters. This can be accomplished by coercing the ATPG vectors into closely matched processor instructions and register file values, with the goal of creating a small binary program that provides similar levels of fault coverage. The executable is run at maximum processor speed and is engineered with tests that target delay faults as a means of maximizing coverage of wear-out related failures. Once constructed, the fault-free counter values obtained from a fault-free run of this specialized executable can be stored in a non-volatile memory or shadow registers, along with the engineered executable. Reliability monitoring can then be carried out periodically by running the engineered executable at high frequency in the field and comparing the measured counter values with the stored values. In contrast to the on-line method proposed in [7], this strategy utilizes existing instructions, requires no scan chain nor major changes to the microprocessor architecture.

A continuous monitoring, program-independent approach is also possible but requires the identification of a set of architecture-specific invariants, i.e., a set of fixed relationships between logic nodes in the architecture. An example would be a node with a specific logic value in the first pipeline stage that always produces, under fault-free conditions, a specific value(s) in a down-stream node(s). An instrumented design with simple logic comparators, or counters if predefined check points are defined, would perform self-assertion-based consistency checks between these nodes. The methodology can focus on the identification of invariants that are sensitive to information leakage faults as a means of reducing the complexity of the analysis and to minimize instrumentation overhead. The nodes in Potato's reduction logic networks associated with the branch condition module of the execute pipeline stage will be evaluated in future work to determine if it is possible to construct self-asserting consistency checks that can detect leakage sensitive faults while remaining independent of the executable and input data.

## 5. Conclusions

This paper investigates a counter-based periodic built-in self-test strategy for detecting faults in the Rocket RISC-V microprocessor, using an FPGA emulation platform. The Rocket design is synthesized using a standard cell ASIC CAD tool flow and an instrumented design is created in which fault-injection and counter circuits are inserted in series with all gate inputs in the netlist. Fault injection experiments are carried out in which one of four fault types is activated on each of the 85,714 nodes within Rocket, one-at-a-time, and as the microprocessor executes the AES algorithm. After each emulation, the count values associated with the 85,714 counters are scanned out and compared with the fault-free values to determine the latency-to-detection of the unmasked (active) faults.

The detection and latency capabilities of the counter-based approach are evaluated on a subset of the active faults referred to as severe faults. The severe faults are defined as faults that leak sensitive information, e.g., a portion of the plaintext and/or encryption key, on the serial port output. A set of five counters, identified as providing the highest fault coverage for the complete set of active (unmasked) faults (called All-faults), are also found to provide complete coverage of all severe faults. Moreover, these counters, individually and as a group, detect all severe faults with latencies less than the occurrence of leakage on the serial port. Although our experiments determined that it is possible to detect the severe faults at even lower latencies using an optimal set of 99 counters, the overhead is much larger than utilizing one or more of the five All-fault counters in a practical RISC-V application. Moreover, each of the five All-faults counters maximizes collateral fault coverage (more than 90% for each counter) on the complete set of faults in the All-faults class, providing an additional benefit beyond detecting all severe faults.

**Author Contributions:** These authors contributed equally to this work. Conceptualization, J.P. and B.D.; methodology, B.D., J.P., D.E.O.J. and T.J.M.; software, J.P. and J.J.; validation, B.D. and J.P.; formal analysis, D.E.O.J.; investigation, B.D., J.P. and T.J.M.; resources, J.P.; data curation, J.P.; writing—original draft preparation, J.P.; writing—review and editing, D.E.O.J., T.J.M. and B.D.; visualization, J.P.; supervision, J.P., T.J.M. and B.D.; project administration, B.D. and T.J.M.; funding acquisition, B.D. and T.J.M. All authors have read and agreed to the published version of the manuscript.

**Funding:** Sandia National Laboratories is a multimission laboratory managed and operated by National Technology & Engineering Solutions of Sandia, LLC, a wholly owned subsidiary of Honeywell International Inc., for the U.S. Department of Energy's National Nuclear Security Administration under contract DE-NA0003525. This paper describes objective technical results and analysis. Any subjective views or opinions that might be expressed in the paper do not necessarily represent the views of the U.S. Department of Energy or the United States Government. SAND2022-10390 J.

**Institutional Review Board Statement:** Not applicable.

**Informed Consent Statement:** Not Applicable.

**Data Availability Statement:** Not Applicable.

**Conflicts of Interest:** The authors declare no conflict of interest.

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
