# Peer review of "Node Monitoring as a Fault Detection Countermeasure against Information Leakage within a RISC-V Microprocessor"

_cryptography, doi:10.3390/cryptography6030038_

Round 1

Reviewer 1 Report

This paper is very well executed. The work is based on idea of keeping track of node toggle activity in counters. Comparing the counter values under fault with the reference ones for fault free execution abnormal behavior can be detected in a periodic testing setting. This paper extends earlier work with a minimisation of the number of counters that are needed for sufficiently good results.

paper positioning:

The proposed technique should work for periodic testing with a gold reference program, input and hence behavior. How this expends to coverage is something that is not addressed in the paper.

As work pertains to periodic testing, how does it fare against other such works? It may very well be that this is a valid, competitive, or better approach, but this is not addressed in the paper.

* the paper spin is "against information leakage". This is largely undefined in the paper. Some faults are claimed to leak information in a particular board, with a particular program, and potentially with specific inputs that are evaluated. The entire evaluation is custom to a single purpose application. 

Authors should investigate if their results generalise: (a) against many inputs (as all would be infeasible), and (b) against other programs/algorithms. The claim that other encryption algorithms are similar and hence these counters would work is valid only if the counters have architectural significance. If so, why not approach the problem systematically from the (micro)architecture and get better "coverage" across algorithms?

there is an implicit assumption that increased toggle rate leads to earlier transistor wear and failure. This is way simplistic. If it were true, the clock and FFs would fail first.

Information leakage is stated as key to this work, but no details are offered. A possible interpretation is that in the previous work you identified the "severe" faults (no definition/explanation here), and you just focus on these. So this work contribution is not against  information leakage, but just an optimisation of testing cost against a specific set of faults. What is the state of the art in this direction?

Why do you execute ASIC  P&R all the way to layout and then go back to netlist? What are the effects (and presumably benefits) of this approach?

The FPGA accelerated framework appears very similar to the one in the authors previous work. If there are significant changes here they should be explicitly identified.

Author Response

Comment 1.1 This paper is very well executed. The work is based on idea of keeping track of node toggle activity in counters. Comparing the counter values under fault with the reference ones for fault free execution abnormal behavior can be detected in a periodic testing setting.

Reply: Thank you

Comment 1.2 — This paper extends earlier work with a minimisation of the number of counters that are needed for sufficiently good results.

Reply: We apologize for not making this clear. Our previous work was focused on building a fast fault injection/emulation engine on an FPGA and then identifying a set of leakage faults. The counters are a novel concept in this paper and are not discussed in any of our previous works. We clarified this in the Contributions Section 1.1.

Comment 1.3 - The proposed technique should work for periodic testing with a gold reference program, input and hence behavior. How this expends to coverage is something that is not addressed in the paper.

Reply: We report complete fault coverage for stuck-at, delay and invert faults in Section 4.2 (the All-faults class) to address coverage beyond leakage fault coverage. We are not sure if this addresses your concern however.

Comment 1.4 - As work pertains to periodic testing, how does it fare against other such works? It may very well be that this is a valid, competitive, or better approach, but this is not addressed in the paper.

Reply: We have not found other works that specifically target faults that result in leakage events. And as we show, our countermeasure is lighter weight because we constrain the fault detection objective to a much smaller of faults. Therefore, it is difficult, and somewhat unfair, to compare our CM approach with others that target a full set of faults. We have added clarification and a comparison table showing the most closely related work in Section 2.0 (Related Work). The added paragraphs attempt to provide a better comparison of overhead.

Comment 1.5 - * the paper spin is "against information leakage". This is largely undefined in the paper. Some faults are claimed to leak information in a particular board, with a particular program, and potentially with specific inputs that are evaluated. The entire evaluation is custom to a single purpose application. 

Authors should investigate if their results generalise: (a) against many inputs (as all would be infeasible), and (b) against other programs/algorithms. The claim that other encryption algorithms are similar and hence these counters would work is valid only if the counters have architectural significance. If so, why not approach the problem systematically from the (micro)architecture and get better "coverage" across algorithms?

Reply: We added text to the Introduction that helps clarify our definition of 'leakage-related faults' We have also added a new section (Section 5.4) which gives results on applying this methodology to the Potato RISC-V core as further support that the counters are located on nodes of architectural significance. The potato processor is not auto-generated from Chisel and allowed the connectivity of the top counters to be tracked to a specific functional unit within the microprocessor. Section 5.5 explains the direction we are currently taking, both on extending the periodic counter-based method and on a continuous monitor based approach.

Comment 1.6 - there is an implicit assumption that increased toggle rate leads to earlier transistor wear and failure. This is way simplistic. If it were true, the clock and FFs would fail first.

Reply: We agree this is an over-simplified approach and have removed that section.

Comment 1.7 - Information leakage is stated as key to this work, but no details are offered. A possible interpretation is that in the previous work you identified the "severe" faults (no definition/explanation here), and you just focus on these. So this work contribution is not against  information leakage, but just an optimisation of testing cost against a specific set of faults. What is the state of the art in this direction?

Reply: We have clarified the definition of leakage faults in the Introduction and in the contribution section (Section 1.1). As we discuss in Section 2 (Related work), we have not be able to find any related work on detecting this specific class of faults.

Comment 1.8 - Why do you execute ASIC P&R all the way to layout and then go back to netlist? What are the effects (and presumably benefits) of this approach?

Reply: We want to emulate a version of Rocket on the FPGA that represents the best possible match to the structural characteristics that would be present if Rocket were built as a hardwired, dedicated integrated circuit (we do not propose to run Rocket as a soft microprocessor on an FPGA -- the FPGA is only used for emulating and studying Rocket with faults and counter circuits introduced.) The proposed CAD tool flow is the flow that would be used to build the hardwired version of Rocket. Once we get to the layout representation, we need to extract a netlist description of this layout to use as input to the emulated implementation on the FPGA. We prevent the FPGA tool Vivado from performing optimizations as well, so the circuit emulated in the FPGA is structurally identical to the version that would be build as an ASIC. We have further clarified these goals in Section 3.5 of the paper.

Comment 1.9 - The FPGA accelerated framework appears very similar to the one in the authors previous work. If there are significant changes here they should be explicitly identified.

Reply: Thank you for pointing this out. The FIM has been enhanced to carry out a binary search mechanism to quickly determine latency, which required a lot of changes when it was all said-and-done. We have modified the bullet in the contributions, Section 1.1, to emphasize this enhancement.

Reviewer 2 Report

Summary: This paper presents a study on how to use a simple counter-based detection method to detect fault injections in RISC-V. 

Strengths:

  1. The authors performed extensive experiments and clearly documented the process and the results in the paper.
  2. The authors investigated latency of detection, which is usually overlooked in existing works.

Weaknesses:

  1. The authors did their case study on the AES encryption algorithm. Since RISC-V is a general-purpose processor, I would like to know how general/effective the proposed counter-based detection method will be on other cryptographic/non-cryptographic algorithms.
  2. Based on the experiments, the authors identified five counters which provide the highest fault coverage. How general is this result when we consider other applications besides AES?
  3. Missing reference on line 94.

Author Response

Comment 2.1 - Strengths:
    The authors performed extensive experiments and clearly documented the process and the results in the paper.
    The authors investigated latency of detection, which is usually overlooked in existing works.

Reply: Thank you.

Comment 2.2 - Weaknesses:
    The authors did their case study on the AES encryption algorithm. Since RISC-V is a general-purpose processor, I would like to know how general/effective the proposed counter-based detection method will be on other cryptographic/non-cryptographic algorithms.

Reply: Although we appreciate your question, our primary goal is to detect key and plaintext leakage in the AES algorithm in this work. As further support of the approach, we added a new section (Section 5.4) which shows similar results when applied to a different architecture, in particular the Potato RISC-V core.

Comment 2.3 -  Based on the experiments, the authors identified five counters which provide the highest fault coverage. How general is this result when we consider other applications besides AES?

Reply: Again, we appreciate your question. Section 5.4 describes our on-going work that is focused on generalizing the methodology. The present work is focused on presenting a novel methodology and providing an analysis that demonstrates proof-of-concept. We were able to add additional evidence and more clarity related to why the approach is effective by applying the methodology to a second microprocessor in Section 5.4.

Comment 2.4 - Missing reference on line 94.

Reply: Thank you. We fixed the reference.

Reviewer 3 Report

This paper provides a counter-based periodic testing strategy for detecting faults in the Rocket RISC-V microprocessor using an FPGA emulation platform. The idea is very interesting. However, the paper has some major revisions that need to be addressed before the acceptance for publication. 

The structure of the abstract is not professional. The be a speech about the counters is repeated several times. The abstract should be written as: 

Introduction- research gap/motivation - problem statement - proposed solution/methodology - results and comparisons - conclusions.

The authors should not use any abbreviation before defining it for the first use. Please check all the papers and make sure all abbreviations are defined at the first appearance.

In the introduction, the authors mentioned that "In the context of a microprocessor, applications that encrypt data are very popular, but are prone to leak secret keys and plaintext when failures occur." Of course, this statement is very general and not exactly right. Such generalization should, indeed, be supported by examples. Please re-write in a more precise way. Also, the paper has several other generalized statements. Please check, revise, and highlight them.

The literature review section should provide a table that summarizes the surveyed papers in this paper and then to gain more insights and come up with a clear justification for the proposed solution approach.

Since the latency of the system is measured and reported as a measure of clock cycles, it's also recommended to provide an analysis of the critical path delay in terms of seconds.

The authors need to analyze and report the area of the design. This can include The number of logic elements (LEs), The number of four-input LUTs, The number of registers, Memory utilization, Total number of I/O pins, and I/O utilization.

The authors need to analyze and report the power consumption of the design. This can include Static thermal power dissipation, Dynamic thermal power dissipation, and Total FPGA thermal power dissipation.

All results regarding the proposed model should be summarized and reported in one table.

Results and Data Discussion is a poor section and not well presented. This section is the core section of your article. the comparisons with another state of the arts are crucial.

Author Response

Comment 3.1 - The structure of the abstract is not professional. The be a speech about the counters is repeated several times. The abstract should be written as: 

Introduction- research gap/motivation - problem statement - proposed solution/methodology - results and comparisons - conclusions.

Reply: Thanks for pointing this out. We have re-written the first half of the abstract to specify the missing motivation, research gap and problem statement components. 

Comment 3.2 - The authors should not use any abbreviation before defining it for the first use. Please check all the papers and make sure all abbreviations are defined at the first appearance.

Reply: We have re-read the manuscipt carefully and hopefully have fixed all of these.

Comment 3.3 - In the introduction, the authors mentioned that "In the context of a microprocessor, applications that encrypt data are very popular, but are prone to leak secret keys and plaintext when failures occur." Of course, this statement is very general and not exactly right. Such generalization should, indeed, be supported by examples. Please re-write in a more precise way. Also, the paper has several other generalized statements. Please check, revise, and highlight them.

Reply: We agree that the original text in the first paragraph of the introduction was too general and have re-written it to be more precise.

Comment 3.4 - The literature review section should provide a table that summarizes the surveyed papers in this paper and then to gain more insights and come up with a clear justification for the proposed solution approach.

Reply: We added a comparison table and have expanded on the discussion of the Related work, Section 2.

Comment 3.5 - Since the latency of the system is measured and reported as a measure of clock cycles, it's also recommended to provide an analysis of the critical path delay in terms of seconds.

Reply: Although we agree this is a good question, it is not meaningful to attempt to answer questions about delay in an emulated environment. Although the methodology followed to create the FPGA emulated version maintains a structural equivalence to an ASIC version, it is not a good model for representing the delay characteristics of the ASIC version. Also, nearly all of the related work on the performance impact associated with periodic and continuous symptom monitors focuses on the time taken to do the fault detection. We have taken a similar approach to explaining the performance overhead in Section 2 at the end of Related work. 

Comment 3.6 - The authors need to analyze and report the area of the design. This can include The number of logic elements (LEs), The number of four-input LUTs, The number of registers, Memory utilization, Total number of I/O pins, and I/O utilization.

Reply: The area overhead of the counter-based CMs, and design area are now reported at the end of Section 2 (Related work) for the ASIC version of the design. We've also added an FPGA resource utilization section to Section 3.5.

Comment 3.7 - The authors need to analyze and report the power consumption of the design. This can include Static thermal power dissipation, Dynamic thermal power dissipation, and Total FPGA thermal power dissipation.

Reply: Although your comment is well-taken, it is not meaningful to measure and report power consumption in the emulated FPGA environment. The FPGA power consumption, instrumented with fault injection circuits and a full set of counters, will not well reflect the power consumption in an actual ASIC version of the microprocessor outfitted with only a couple counter circuits. Moreover, given we are proposing only a small number of counters, their power consumption will be negligible in comparison to that of entire core.

Comment 3.8 - All results regarding the proposed model should be summarized and reported in one table. Results and Data Discussion is a poor section and not well presented. This section is the core section of your article. the comparisons with another state of the arts are crucial.

Reply: We have included a comparison in Related work and have added an itemized list at the end of Section 4.3 the summarizes the main results from the experiments.

Round 2

Reviewer 1 Report

The paper has  improved in this round as the authors did a fairly good job with the reviewer comments.

My main concern that remains from the first round is the narrow scope of the approach. The authors optimise a fault detection technique with a particular (SW) application in mind. Their solution is indeed smaller than competing techniques, but these techniques attempt a global approach that detect (ideally) all faults. 

The very focus of the work prevents it from being comparable with other previous works. 

That being said, the authors ported their technique on a different processor implementation, which shows that their results (as narrow as they are) are not specific to a particular implementation.

In terms of presentation, (§2) I would expect that the discussion would start from the system in mind and how one can exploit it, and then go to protection/mitigation techniques. Instead the paper dives in details about the fault injection campaign and FPGA implementations. However, in my understanding the work addresses a general technique applicable to ASICs. The presentation as is confuses the issue.

The abstract should be re-written as it does not describe accurately what the paper is about

Author Response

Comment:

... The very focus of the work prevents it from being comparable with other previous works. 

Reply:

Agreed -- we have elaborated on this difficulty in the new Section 4.5.

Comment:

...  However, in my understanding the work addresses a general technique applicable to ASICs. The presentation as is confuses the issue.

Reply:

We have added clarifications to Section 1.1, and an introductory paragraph to Section 3.0. 

Comment:

The abstract should be re-written as it does not describe accurately what the paper is about

Reply:

Please see the highlighted section of the abstract, which has been re-written.

Reviewer 3 Report

Even though the authors have NOT responded to all of my comments, however, I can say that the amount of contribution in this paper is enough for publication. Just one recommendation is to move Table 1 along with its text to  the end of the results section.

Author Response

We have moved the overhead comparison text and table to a new Section 4.5.